# Training vs. Tolerance: The Yin/Yang of the Innate Immune System

**DOI:** 10.3390/biomedicines11030766

**Published:** 2023-03-02

**Authors:** Trim Lajqi, Natascha Köstlin-Gille, Reinhard Bauer, Sotirios G. Zarogiannis, Esra Lajqi, Valdrina Ajeti, Stefanie Dietz, Simon A. Kranig, Jessica Rühle, Ardian Demaj, Janine Hebel, Maria Bartosova, David Frommhold, Hannes Hudalla, Christian Gille

**Affiliations:** 1Department of Neonatology, Heidelberg University Children’s Hospital, D-69120 Heidelberg, Germany; 2Department of Neonatology, University of Tübingen, D-72076 Tübingen, Germany; 3Institute of Molecular Cell Biology, Jena University Hospital, D-07745 Jena, Germany; 4Department of Physiology, School of Health Sciences, Faculty of Medicine, University of Thessaly, GR-41500 Larissa, Greece; 5Department of Radiation Oncology, Heidelberg University Hospital, D-69120 Heidelberg, Germany; 6Department of Pharmacy, Alma Mater Europaea—Campus College Rezonanca, XK-10000 Pristina, Kosovo; 7Faculty of Medical Sciences, University of Tetovo, MK-1200 Tetova, North Macedonia; 8Center for Pediatric and Adolescent Medicine Heidelberg, University of Heidelberg, D-69120 Heidelberg, Germany; 9Klinik für Kinderheilkunde und Jugendmedizin, D-87700 Memmingen, Germany

**Keywords:** trained immunity, tolerance, inflammatory, dose, PAMPs, DAMPs, epigenetic, metabolic, signaling, diseases

## Abstract

For almost nearly a century, memory functions have been attributed only to acquired immune cells. Lately, this paradigm has been challenged by an increasing number of studies revealing that innate immune cells are capable of exhibiting memory-like features resulting in increased responsiveness to subsequent challenges, a process known as trained immunity (known also as innate memory). In contrast, the refractory state of endotoxin tolerance has been defined as an immunosuppressive state of myeloid cells portrayed by a significant reduction in the inflammatory capacity. Both training as well tolerance as adaptive features are reported to be accompanied by epigenetic and metabolic alterations occurring in cells. While training conveys proper protection against secondary infections, the induction of endotoxin tolerance promotes repairing mechanisms in the cells. Consequently, the inappropriate induction of these adaptive cues may trigger maladaptive effects, promoting an increased susceptibility to secondary infections—tolerance, or contribute to the progression of the inflammatory disorder—trained immunity. This review aims at the discussion of these opposing manners of innate immune and non-immune cells, describing the molecular, metabolic and epigenetic mechanisms involved and interpreting the clinical implications in various inflammatory pathologies.

## 1. Introduction

In the early 20th century, the Nobel laureate Paul Ehrlich, in a spectacular lecture, introduced the concept of specific antigen–antibody interaction, which established the fundamental basis of the adaptive arm of immune responses. Later, several reports clarified major questions related to adaptive (acquired) immunity, which relies mainly on two different types of lymphocytes, T and B cells, able to recognize pathogen-specific antigens and develop immunologic memory against them whilst promoting the long-term specific elimination of the pathogen after a subsequent encounter [1,2].

Different from the acquired immune system, innate immune cells were considered as evolutionary archaic due to the lack of memory responses, characterized particularly by a rapid cell recruitment to the site of infection, producing various inflammatory mediators (i.e., cytokines and chemokines), and by engulfing the pathogen, thus eliminating the infectious invader [3,4]. The overall impression that immunological memory was an exclusive hallmark of the adaptive immune response was challenged by several studies in plants and invertebrates, indicating a greater protection against reinfection [5,6,7]. Interestingly, since the last universal common ancestor of all cells (LUCA) evolved into diverse phylogenic groups where the majority (>95%) of the living organisms relies on the protection responses driven mainly by the innate immune system, the acquisition of adaptive-like immune memory for innate immune cells may be considered as an evolutionary success [7,8,9,10]. Initial observations about memory effects arising from the innate immune system were reported in 1934 by a Swedish scientist Naeslund, then followed by a report of Mackaness (in 1968) and later from Garly et al. (in 2003), showing that mammals vaccinated by Bacillus Calmette–Guerin (BCG) improved survival rates with effects exceeding the burden of tuberculosis, affording appropriate non-specific protection against various infectious diseases [11,12,13,14]. Similar findings about the non-specific protective effects of BCG as well as measles were reported by other groups [15,16,17,18,19].

Netea and colleagues, in a seminal report in 2011, suggested the concept of trained immunity, defining it as a non-specific property of innate immune cells as well as tissue-resident stem cells to act against secondary infectious challenges [20,21]. Trained immunity, known also as innate memory or trained innate memory (abbr. TRIM), describes a long-term functional reprogramming evoked by various endogenous danger signals (damage-associated molecular patterns, DAMPs) released upon cellular stress or tissue injury, or exogenous pathogenic conserved molecules (pathogen-associated molecular patterns, PAMPs) [21,22,23,24]. Notably, recent studies have demonstrated also the capability of hormones to support the induction of trained immunity, especially by sex-related (i.e., β-estradiol) hormones or aldosterone as a mineralocorticoid that regulates electrolyte homeostasis and affects many cardiovascular inflammatory events [25,26].

Trained cells adopt a prolonged activated phenotype characterized by enhanced levels of pro-inflammatory mediators ( such as IL-1β, TNF-α, IL-6 and reactive oxygen species (ROS)) and increased antimicrobial and antitumoral activities, a process regulated mainly by epigenetic changes with resulting changes in metabolism and their functional reprogramming [27,28]. In contrast to trained immunity, endotoxin tolerance (ET) as an opposing feature is well known and represents a transient unresponsive state of immune cells against further challenges characterized by decreased pro-inflammatory signatures and an increased anti-inflammatory mode [29,30]. Chromatin modifications and gene reprogramming supported by metabolic changes have been demonstrated as crucial developments supporting ET [30,31,32,33,34,35,36,37]. Innate immune cells sense different PAMPs and DAMPs by germline-encoded pattern recognition receptors (PRRs), resulting in downstream signaling events culminating in the release of various inflammatory mediators [38,39,40]. To date, training as well as tolerance reactions are mainly imposed by alterations occurring in different classes of PRRs, such as Toll-like receptors (TLRs), nucleotide oligomerization domain (NOD)-like receptors (NLRs), retinoic acid-inducible gene-I (RIG-I)-like receptors (RLRs) and C-type lectin receptors (CLRs) instructing local and systemic inflammatory responses in the living organisms [40,41,42,43]. Altogether, the activation of these signaling pathways contributes to the evolvement of inflammatory diseases. To date, an increasing number of studies has shown the clinical importance of trained immunity as well as tolerance implicated in numerous inflammatory diseases [21,23,44,45,46,47,48].

Trained innate immunity as well as the topic of endotoxin tolerance have gained considerable interest through the years (Figure 1). In the present review, we discuss in depth the different characteristic properties of trained immunity and endotoxin tolerance and their role in various inflammatory diseases.

## 2. Trained Immunity

Trained immunity represents a functional state of the innate immune cells and tissue-resident stem cells adjusting their response to subsequent insults, characterized particularly by a prolonged activation after a resting period, orchestrated mainly by epigenetic reprogramming and the metabolic rewiring of cells [24]. It describes a property of innate immune system to mount memory-like responses against past microbial and non-microbial challenges [20,21,24]. To date, studies have shown that trained immunity can be maintained in innate immune cells for months up to decades, a process occurring in the bone marrow progenitor cells–central trained immunity, or for days up to a week in circulating innate immune cells–peripheral training [21,49,50,51,52].

Emerging investigations have revealed two distinct hypotheses about the induction of trained immunity and tolerance in innate immune cells: first, the stressor-dependent hypothesis, where priming by specific PAMPs/DAMPs (i.e., β-glucan, BCG, oxidized low-density lipoprotein (oxLDL) and heme) triggers the induction of training, whereas the Gram-negative endotoxin (i.e., lipopolysaccharide (LPS)) promotes only tolerant reactions; and the dose-dependent hypothesis, which demonstrates a biphasic dose–response where low-dose priming results in a trained phenotype, whereas high-dose exposure triggers an immune-suppressive phenotype (tolerance) after a secondary insult [20,21,53,54]. Moreover, other reports elaborated also the role of corticosteroid hormones, dietary restriction and maturation state supporting the induction of these adaptive cues in innate immune cells.

### 2.1. Stressor-Dependent Induction of Trained Innate Memory

Innate immune cells as gatekeepers recognize various pathogens, mounting immediate immune resistance reactions aimed at eliminating the microbial intruder [53]. In the early 20th century, Shwartzman described long-term adaptation reactions occurring by continuous intravenous injections of the same filtrate promoting hemorrhagic necrosis at the site of injection in the skin [55]. This phenomenon, where cells sense the initial injection supporting resistance mechanisms against a secondary challenge, is known as Shwartzman reaction.

Later, Netea and colleagues proposed the concept of stressor-dependent induction of trained immunity in innate immune system [20]. They principally showed that the pre-exposure of monocytes to the fungal-cell-wall-constituent β-glucan (*C. albicans*) triggers an exaggerated burst of inflammatory responses upon secondary infectious stimuli (i.e., LPS), a process called trained immunity [56,57]. Further in vitro and in vivo studies demonstrated comparable effects when innate immune cells (monocytes, macrophages, microglia, neutrophils, mast cells and dendritic cells) as well as hematopoietic stem cells, primed by β-glucan or *Candida albicans* infections, promote protective reactions against various infections (such as *Mycobacterium tuberculosis*, *Leishmania braziliensis* and influenza) or support anti-tumoral effects in a ROS- or CCR2-dependent manner [52,58,59,60,61,62,63,64]. *C. albicans* as well as β-glucan induce a functional reprogramming of innate immune cells, resulting in enhanced release of pro-inflammatory mediators (i.e., IL-1β, TNF-α, IL-6, IL-32 and ROS) mediated by an IL-1-dependent signaling [58,59,62]. Similarly, LPS guides the induction of training or tolerance depending on the type of secondary stimulus [65].

Extensive studies from different research groups have led to the identification of other inducers of trained immunity. Muramyl dipeptide (MDPs) and tripeptide (MTPs), as bacterial wall peptidoglycans recognized by NOD2 receptors, were reported to induce trained immunity and inhibit cancer growth by sensitizing melanoma cells to inhibitory checkpoint therapy [66,67]. Likewise, *Pseudomonas aeruginosa* serves as an important pathogen-associated microbial component, causing acute and chronic lung infections [68,69]. Bigot et al., in a proof-of-concept study, demonstrated the ability of human respiratory epithelial cells to memorize former exposures to *P. aeruginosa* and provide protective reactions against a wide spectrum of pathogens, a process relying on epigenetic regulation [70]. Similarly, in another investigation, where mice were trained by a wide range of infections such as *S. aureus*, *L. monocytogenes*, *E. coli*, *C. rodentium* and *P. aeruginosa* to mimic systemic infections, peritonitis, enteritis and pneumonia, the ability of mice to exhibit wide-range protective effects from heterologous infections at distant anatomic sites from the training site, particularly regulated by IL-1 pathway, was revealed [71,72]. The data presented by these studies may be of crucial importance since they provide a double-edge property of trained immunity: first, providing antimicrobial responses against heterologous infections, and second, serving as a therapeutic tool to control inflammatory exacerbations during multi-microbial infections. However, these findings should be considered with caution since *P. aeruginosa*, during pulmonary infections, might utilize the host defense system to its advantage by redirecting the host metabolism toward itaconate production, promoting biofilm formation resulting in human tissue colonization [73]. Extensive studies are required to further determine all the detrimental factors (i.e., epigenetic regulation, metabolic rewiring and former exposures) that can promote trained immunity and further elucidate the clinical relevance.

To date, it is well-known that BCG vaccination is associated with decreased mortality rates, driven mainly by its non-specific protective actions against other infections in addition to the targeted pathogens [14,74,75,76]. Furthermore, it has been hypothesized that these non-specific protective effects of certain vaccines, such as BCG, measles and polio, are driven by the innate immune system. Kleinnijenhuis and colleagues provided firm evidence that BCG promotes the non-specific long-term boosting of innate immune system, trained immunity, mediated by a NOD2-dependent regulation of epigenetic marks such as histone H3 at lysine 4 (H3K4) trimethylation [77,78]. γδ T cells, as a unique T-cell subpopulation featuring functions of both immune systems, innate and adaptive, not restricted to the major histocompatibility complex (MHC) of signaling, represent a key role for the innate immune system of responses to various microbial infections [79,80]. A growing body of evidence demonstrated that γδ T cells are early responders sensing different viruses by their PRRs, natural killer type receptors (NKRs) or via T-cell receptors (TCRs) exerting their protective function by a rapid and effective resolution of the viral infection [80,81,82,83,84,85]. In a recent work, Röring et al., elaborated on the role of the measles, mumps and rubella (MMR) vaccine, affording a broader protection against heterologous infections, and found that MMR triggers profound long-term functional changes (trained immunity) in γδ T cells characterized by increased levels of TNF-α and IFN-γ as well as an enhanced reliance on mitochondrial metabolism [86]. Moreover, several studies have reported the analogous contribution of BCG vaccination to γδ T cells expressing a typical heterologous effect observed in trained immunity [87,88,89,90].

In a recent report, Debisarun et al., demonstrated that the quadrivalent inactivated influenza vaccine triggers the transcriptional reprogramming of innate immune cells, training effects, resulting in fine-tuned anti-SARS-CoV-2 responses supporting enhanced protective reactions against COVID-19 [91]. Moreover, a study indicates that a single dose of the ChAdOx1 nCoV-19 vaccine (AZD1222), which uses an adenoviral vector, displayed a prolonged effective protection by myeloid cells characterized by heightened levels of pro- and anti-inflammatory cytokines (i.e., IL-1β, IL-6, CXCL1 and IL-10) in response to subsequent unrelated stimuli, featuring innate memory cues [92]. Other attempts included the investigation of viral components promoting innate memory. Comparable effects have been shown also by murine infections with cytomegalovirus (CMV) displaying a T/B-cell-independent and natural killer cell (NK)-dependent protective mechanism against re-infections [93,94]. In addition, earlier studies related to NK cells revealed a unique property of this type of cells expressing memory-like features driven by previous encounters to CMV infection or mediated by a cocktail of inflammatory cytokines (IL-12, IL-15 and IL-18) [95,96,97,98]. Both, murine and human CMV promoted the antigen-specific induction of memory NKs [93,98,99,100]. Nevertheless, cytokine-mediated memory NKs expressed non-specific adaptive manners with enhanced proliferative capacity, elevated interferon-γ (IFN-γ) and prolonged persistence [98,101,102,103]. Because of these adaptive cues, NK cells can be considered as an evolutionary bridge between acquired and innate immune system.

Different ventures of existing vaccines or viral components to induce trained immunity seek to improve conventional vaccines toward the new formulations of the so-called trained immunity-based vaccines (TIbV), aimed at stimulating broader responses beyond their specific antigens, an approach that could make it possible to afford proper resistance against viral outbreaks [104,105,106,107]. However, this might not be the case for allergen vaccines. In a recent study Benito-Villalvilla et al., it was demonstrated that allergoid–mannan conjugates, which serve as next-generation vaccines for allergen-specific immunotherapy (AIT), contribute to allergen tolerance by the differentiation of monocytes into tolerogenic dendritic cells (DCs), but lack the capacity to induce any trained immunity effects [107,108]. Similarly, the innate counterparts of T cells—innate lymphoid cells (ILCs)—primed by IL-33 or allergic inflammation mount enduring memory-like responses against other allergens, which can serve as double-edged sword, on the one side affording proper protection against infectious diseases and, on the other side, by contributing to pathology or disease progression [109,110,111]. Altogether, these data indicate that further research is necessary to clarify the role of different vaccines, allergens or cytokines promoting training or tolerant reactions.

The multifold interplay between microbiota and the immune system during homeostasis or diseases represents a crucial factor orchestrating continuously the development of host’s immune responses whilst maintaining its microbiota composition [54,112,113]. The probiotic gut commensal *Limosilactobacillus reuteri* recalls enhanced responses with mixed secondary responses in human monocyte-derived DCs upon subsequent exposure, accompanied with enriched histone modifications and portrayed particularly by an increased IL-1β and IL-6 phenotype, but diminished TNF-α, IL-23 and IL-27 [114]. However, blood DCs primed by *L. reuteri* exclusively resembled a tolerant phenotype upon a second challenge. Furthermore, our group demonstrated that gut microbiota cell-derived nano-sized biovesicles, known as extracellular vesicles (EVs), drive the induction of innate memory in murine bone-marrow neutrophils in a TLR2-dependent manner in vitro [115]. Similarly, the role of extracellular vesicles from HIV-infected humans maintaining persistent chronic inflammatory responses (phenomenon of trained immunity) of myeloid cells, associated with increased morbidities against secondary infections, has been described by Dubrovsky et al. [116]. Although there are still many issues to be clarified, these studies nonetheless shed some encouraging light to the role of microbiota or bilipid-enclosed envelopes, such as EVs, encouraging trained effects in innate immune cells.

In addition to the aforementioned PAMPs, DAMPs, such as oxLDL, heme, urate, vimentin and high-mobility group box 1 (HMGB1), recognized mainly by TLR4, can be targeted for trained immunity modulation [24,117,118]. Pre-exposure to specific DAMPs as non-microbial triggers can induce the functional reprogramming of innate immune cells triggering an altered immune response toward a subsequent challenge [117]. DAMPs released during tissue injury from cell death or after organ transplantation allow innate immune cells to elicit long-term epigenetic changes at the promoters of inflammatory genes and cause metabolic rewiring [24,119]. Vimentin and HMGB1 represent inflammatory mediators upregulated during organ transplantation, responsible for acute and chronic transplant rejection [120,121]. A study by Braza et al., demonstrated that vimentin and HMGB1 drive the trained immunity-associated cytokine production of infiltrating macrophages during allograft rejection [23]. Moreover, a large number of infiltrating innate immune cells, especially macrophages and neutrophils during transplant rejection, present a significantly higher phagocytic activity if exposed to HMGB1 [122,123,124]. Another well-described consequence of oxidative stress, oxLDL, is associated with an increased probability of graft rejection [125,126,127,128]. Patients with chronic kidney disease (CKD) display higher serum levels of oxLDL, characterized by an hyperinflammatory phenotype of monocytes accompanied by an increased production of pro-inflammatory cytokines (i.e., TNF-α, IL-1β and IL-6) [128,129,130,131]. Priming by oxLDL, upon secondary exposure to TLR2 and TLR4 agonists, resulted in a training phenotype of macrophages characterized by increased production of inflammatory cytokines, monocyte chemoattractant protein 1 (MCP-1) and matrix metalloproteinase 2 (MMP2) and 9 (MMP9) [117,132]. Likewise, human non-immune cells from the vascular wall primed by oxLDL exhibit pro-inflammatory features regulated presumably by trained immunity effects [133]. Although it is well-known that oxLDL induces trained immunity in myeloid cells by shifting the metabolism toward glycolysis, there are still no data about the role of training in CKD or dialysis patients [24].

Hyperuricemia represents one of the main factors linked to gouty arthritis, caused by elevated levels of uric acid (urate) [134,135,136]. Existing evidence demonstrated that urate induces transcriptional and epigenetic modifications in circulating monocytes resulting in enhanced responsiveness marked by increased levels of IL-1β, indicating a crucial role of trained immunity [136,137,138]. Moreover, heme is another DAMP involved in multiple biological processes that regulate trained responses of innate immune cells by increasing the frequency of myeloid-biased multipotent progenitors, associated with increased monocyte and neutrophil infiltration in the murine peritoneum [139,140].

Mechanistically trained immunity is reported to be supported by different alterations occurring on the PRRs, driving the activation of the mechanistic target of rapamycin (mTOR) through phosphoinositide 3-kinases (PI3Ks) triggering downstream signaling events promoting the activation of nuclear factor-κB (NF-κB), resulting in the production of several inflammatory mediators [21,59,117,141]. Cheng et al., identified a canonical hypoxia sensing system required for the induction of memory-like responses in monocytes, where dectin-1-dependent activation of mTOR by β-glucan provokes increased expression of hypoxia-inducible factor 1α (HIF-1α) [142]. Ongoing studies showed that the downstream activation of intermediate proteins, such as protein kinase B (Akt), by PI3Ks, encouraged by TLR4 activation serves as crucial factor for the induction of innate memory in murine microglial cells [46,143]. Similarly, a recent work elaborated the role of trained immunity driving the pathological inflammation in a murine model of Duchenne muscular dystrophy (DMD), a genetic disease affecting more commonly males than females, revealing TLR4-dependent activation of bone-marrow-derived macrophages as the central cause for the DMD pathogenesis [144]. Furthermore, the activation of PRRs, notably dectin-1 by β-glucan or TLR4 by LPS, triggered the activation of mitogen-activated protein kinase (MAPK)-mediated pro-inflammatory pathway, exclusively extracellular signal-regulated kinase (ERK) 1/2, p38 and Jun N-terminal protein kinase (JNK) supporting the production of pro-inflammatory mediators [56,59,143]. Comparable mechanistic events have been shown also by neutrophils and macrophages primed by lipoteichoic acid (LTA) from the Gram-positive *Staphylococcus aureus* as a TLR2-activator [145,146,147,148].

Epigenetic reprogramming at the promoters of inflammatory genes with resulting changes in cell metabolism have been identified to orchestrate memory-like features in innate immune cells [21,24,28,149]. Epigenetic signatures triggering trained immunity are driven by histone modifications, such as histone methylation and acetylation, resulting in chromatin reconfiguration, rendering a more accessible form of chromatin facilitating the transcription of pro-inflammatory genes [28,117,150]. Current data determined the specific epigenetic hallmarks of trained immunity, including monomethylation or trimethylation of histone 3 lysine 4 (H3K4me1 or H3K4me3) at the promoters of stimulated genes, and the acetylation of histone 3 lysine 27 (H3K27ac) at distal enhancers [21,59,117,151]. MicroRNAs (miRNAs) as small non-coding regulatory RNAs are known epigenetic modulators that can alter the levels of target mRNAs, regulating also the induction of innate memory in innate immune cells [21,152]. Several studies identified miR-155, miR-146a-5p and miR-9-5p to be critical regulators of trained immunity supporting the hyperactivation of myeloid cells [153,154,155,156]. These may likely be a conjoint result of the decreased activity of phosphatases (i.e., SHIP1) by miRNAs [157]. A specific set of long non-coding RNAs (lncRNAs), known as immune gene-priming lncRNAs (IPLs), alters histone modifications, especially H3K4me3 on promoters of pro-inflammatory cytokines supporting trained immunity in macrophages [158]. Moreover, a study suggested that changes occurring in the DNA methylation pattern may discriminate between ‘responders’ and ‘non-responders’, identifying several genes that can be used as predictors of training effects [21,159].

The interplay of numerous intracellular metabolic pathways, such as glycolysis, oxidative phosphorylation (OXPHOS), accumulation of metabolites from the tricarboxylic acid cycle (TCA) and cholesterol biosynthesis, serve as fundamental immune-metabolic circuits endorsing the effects of trained immunity [160,161,162]. The activation of the dectin-1- mTOR- HIF-1α pathway resulted in a shift of metabolism toward aerobic glycolysis with increased glucose consumption, higher levels of L-lactate and elevated levels of its reduced form of nicotinamide adenine dinucleotide (NADH) [142]. The overall notion of glycolysis-induced trained immunity has been challenged by several reports, showing that oxLDL-, LPS-, β-glucan or BCG-trained macrophages adopt OXPHOS as an immunometabolic mechanism to express increased hyperresponsiveness [163,164,165,166]. The accumulation of intermediary metabolites of the TCA cycle, such as fumarate, as a result of glutamine replenishment (glutaminolysis), altogether with cholesterol synthesis contributes to the induction of innate immune memory in myeloid cells, downregulating the activity of KDM5 histone demethylases [167,168]. Moreover, mevalonate as an upstream metabolite of cholesterol synthesis triggers the training of monocytes by an insulin-like growth factor 1 receptor (IGF1-R) and mTOR-dependent signaling pathway, a mechanism that may further explain the hyperactivated phenotype of hyper-immunoglobulin D syndrome (HIDS) pathology [21,169]. Since training effects have been portrayed by enhanced levels of ROS, Ferreira et al., investigated the role of its intracellular molecule scavenger- glutathione, showing a close association of glutathione with the increased production of pro-inflammatory mediators upon trained immunity in human monocytes [170]. Other attempts to understand the role of the pentose phosphate pathway (PPP) in trained immunity responses by the pharmacological blockage of this metabolic event demonstrated this pathway as a dispensable occasion [161,167,171]. Even though, to date, there have been several papers indicating the crucial role of several metabolic pathways supporting the induction of training effects in innate immune cells, individual metabolic variations (i.e., glycolytic variability in diabetes patients) of the organisms should be taken into consideration since they significantly alter this resistance reaction [172].

Neutrophils, monocytes, macrophages, dendritic cells as well as microglial cells as a part of innate immune system are well known to be active phagocytes, producing a variety of antimicrobial substances (i.e., degrading enzymes, antimicrobial peptides and ROS) eliminating invaders [173,174,175]. To date, it has been shown that trained cells display greater antimicrobial efficacy characterized with increased phagocytic activity [50,62,176,177,178]. However, opposing findings were reported for the murine resident macrophages of the central nervous system, microglia, where the enhanced phagocytic capacity was featured exclusively by tolerant microglial cells resulting in decreased accumulation of β-amyloid plaques [46,179,180]. Anti-tumoral effects resulting from trained immunity-induced myelopoiesis caused by epigenetic and transcriptomic rewiring was reported in β-glucan-treated mice manifesting reduced tumor burden in a ROS-dependent manner [61,63]. The increased production of MCP-1 by trained cells triggers altered immune-regulatory properties, specifically elevated cell recruitment and the migration of myeloid cells, maintaining proper immune responses in health and diseases [92,176,181].

Contradictory data have arisen about the intergenerational inheritance of trained immunity providing heterologous resistance against infections in offspring [182,183]. Katzmarski et al., showed that sublethal infections o *C. albicans* or zymosan drive immune-resistance reactions to the progeny against heterologous infections caused by LPS or *E. coli* and *L. monocytogenes* infections [182]. Opposing findings were reported by Kaufmann et al., where the pre-exposure of parental mice to BCG, β-glucan or *C. albicans* did not exhibit any protective responses against viral, bacterial or fungal infections in the offspring [183]. A plausible explanation for this discrepancy of data likely involves several housing or dietary factors, the composition of the microbiome, environmental conditions as well as previous expositions to any infectious stressors [184].

To date, an increasing number of studies has investigated the role of different PRRs, epigenetic marks as well as metabolic changes involved in the induction of training effects, yet many aspects remain obscure. An illustrative demonstration of all the above-mentioned stressor-dependent inducers of trained immunity and their characteristics is presented in Figure 2.

### 2.2. Dose-Dependent Induction of Trained Innate Memory and Immune Tolerance

“The dose makes the poison” (lat. dosis sola facit venenum) was one of quotes credited to Paracelsus underlining the importance of dose exhibiting beneficial or toxic events in organisms. Ongoing studies showed that cells strive to demonstrate hormetic or biphasic dose responses if exposed to a stimuli (intrinsic or extrinsic), where low doses promote stimulatory effects and high doses trigger inhibitory responses [185,186,187]. A similar concept has been proposed for the adaptive responses introduced by innate immune cells upon a secondary insult, where low-dose priming can trigger resistance (training) responses and high-dose priming results in tolerance reactions [53,54,188].

Actually, the role of low-dose priming enhancing the capacity of myeloid cells to subsequent challenges by bacterial infectious constituents was reported in the late 1980s, where the priming of monocytes and macrophages by endogenous IFN-γ upon a secondary stimulus by LPS triggered heightened immune responses [189,190,191], a process regulated particularly by the transcription factor NF-κB [192]. A suitable explanation for this hypothesis may serve the mechanistic interplay of mTOR and AMP-activated protein kinase (AMPK) as central regulators of cell metabolism [193,194,195]. AMPK is known to inhibit mTOR activity, orchestrating a cellular adaptive response promoting catabolic pathways and thus modulating the resistance mechanism toward an immune-suppressed phenotype [196,197]. In contrast, mTOR is well-known to support resistance mechanisms in innate immune cells and promote the memory-like cues of innate immune cells [142,198]. Interestingly, the activation of these signaling pathways depends also on the dose of stressors, where super-low doses promote resistance by a mTOR-dependent mechanism, and tolerant responses are promoted by high-stressor loads mediated by AMPK [53,196,199,200].

A growing body of evidence demonstrated comparable dose-dependent features being responsible for the induction of training and tolerance responses in myeloid cells. The application of low doses of LPS prior to any septic insults resulted in the increased mortality of mice characterized with heightened levels of inflammatory mediators, distinct from high-dose LPS pre-conditioning, which triggered an immune-suppressed phenotype portrayed by diminished levels of TNF-α and reduced sepsis mortality [201]. Furthermore, other in vitro studies revealed similar adaptive dose-dependent responses for LPS in murine monocytes and macrophages, where an increased expression of pro-inflammatory mediators by a low-dose LPS is preferentially dependent on the removal of negative modulators such as B-lymphocyte-induced maturation protein-1 (Blimp-1), different from tolerized cells induced by high doses of LPS supporting an anti-inflammatory phenotype [202,203,204,205]. Likewise, murine bone-marrow neutrophils primed with low doses of Gram-negative (i.e., LPS) or Gram-positive (i.e., LTA) constituents or gut microbiota-derived bacterial EVs upon a secondary stimulus by endotoxin exhibit an increased production of pro-inflammatory cytokines by a TLR4- and TLR-2-dependent manner activation of NF-κB by intermediate signaling of PI3Ks/MAPKs [115,145,206]. In contrast, high-dose priming promoted immune-suppressed anti-inflammatory characteristics with a decreased production of pro-inflammatory cytokines and increased IL-10 [115,145,206]. Mechanistically, the crucial role of NF-κB mediating adaptive manners has been reported for various myeloid cells [143,169,202,204,206]. In humans, a study showed that the increased immune response by monocytic cells relies on the production of IL-6 and IL-12 [207].

Accumulating reports showed that murine microglial cells can express both types of immunological imprinting, training and tolerance, which may impact the pathological hallmarks of neurological diseases [46,143,180,208]. The application of low/single doses of LPS resulted in higher inflammatory responses (trained microglia), exacerbating cerebral β-amyloidosis in a murine model of Alzheimer’s disease [46,143,180]. On the contrary, high/repeated doses of LPS exhibited a tolerant phenotype of microglia supporting neuroprotective effects [46,143,208]. The capability of microglia to retain long-term memory features relies mainly on epigenetic modifications, enrichment (training) or loss (tolerance) of epigenetic marks at enhancer regions [46,143,209,210].

Contradictory findings arise about the role of endotoxin tolerance on the antimicrobial efficacy. While Grondman and colleagues demonstrated that human immunotolerant monocytes display decreased antimicrobial activities ex vivo, other studies revealed contrasting effects, where tolerant myeloid cells are capable to express an elevated antimicrobial capacity characterized by increased phagocytosis [211,212,213,214]. Similar divergent findings have been reported in high-dose primed cells, where murine tolerant microglia expressed increased phagocytic activity in vivo and in vitro, whereas bone-marrow-tolerant neutrophils exhibited diminished phagocytosis in vitro [143,179,206]. This issue needs further investigation in order to clarify under which circumstances immunotolerant cells express increased or decreased killing activities.

As mentioned previously, trained cells express anti-tumoral features, different from immunosuppressive tolerant cells that mimic the M2 phenotype expressing pro-tumoral activities while promoting angiogenesis and tissue repair [215,216,217,218].

A respective illustration of the dose-dependent induction of trained immunity and tolerance response is shown in Figure 3. Since the topic of pathogen dose supporting the induction of opposing adaptive responses is growing, we hypothesize that similar findings will be reported for other PAMPs and DAMPs in several innate immune and non-immune cells.

### 2.3. Ageing, Hormones and Dietary Restrictions as Decisive Modifiers of Innate Memory

The biological mechanism of aging exposes the organism to a high susceptibility to infectious diseases due to an immune-compromised system that is incapable of mounting proper immune reactions [219,220,221,222]. Recent findings have revealed that aging relies on immune responses driven by the innate immune cells characterized by a chronic state of activation, inflammaging, which hampers T-cell recall responses to infections resulting in diminished responses of the acquired immune system [223,224]. To date, the role of innate immune memory by the innate immune compartment during ageing has only been slightly elaborated.

The investigation of human monocytes trained by β-glucan showed that trained cells from aged healthy volunteers exhibit an increased production of the pro-inflammatory TNF-α compared to young healthy individuals [225]. IL-6 was not significantly altered between the groups, while the heightened levels of cell surface markers and TNF-α were associated with an increased acetylation of H3K27 as well as enhanced metabolic capacity [225]. Additionally, as mentioned above, murine microglia as resident innate immune cells in the CNS also mount memory-like responses depending on the pathogen dose [46,143,208]. Different from the human monocytes trained by β-glucan, the trained phenotype of aged microglia induced by low doses of LPS lost the ability to express training effects in vitro [180]. This study demonstrated the pronounced plasticity of newborn microglia to introduce training effects by LPS upon a secondary stimulus. These data may be of relevance to explain the increased neuro-inflammatory susceptibility of the immature brain [180]. Moreover, it is well known that microglia during aging express a so-called “sensitized/primed” state mirrored by a heightened inflammatory profile [221]. High-dose LPS-induced immune tolerance was further significantly pronounced in aged microglial cells in vitro, which can be the result of cells attenuating excessive damage after any recurrent systemic inflammation [180]. This conclusion is also supported by other studies where tolerant microglia promotes neuroprotective reactions in the murine brain [46,208]. However, since the topic of innate memory is quite recent, there are still many aspects of aging that remain unclear and need further investigation.

Aldosterone as a mineralocorticoid hormone is essential for blood pressure and electrolyte homeostasis [226,227]. High levels of aldosterone are associated to plaque formation by monocyte-derived macrophages during atherosclerosis [228,229]. The role of aldosterone promoting training effects has been elaborated lately and the data showed that brief exposure to high amounts of aldosterone in human monocytes upon a secondary challenge by TLR2 and TLR4 ligands in vitro, resulted in enduring memory features with increased levels of pro-inflammatory mediators (TNF-α, IL-6 and ROS) regulated by H3K4me3 modifications occurring at promoters of genes important for the fatty acid synthesis pathway [26]. Therefore, the identification of such relation between supranormal levels of aldosterone and atherosclerosis may be of profound interest to understand the molecular mechanism and for the development of new therapeutic strategies to control the risk of cardiovascular diseases [26,227]. Similarly, sex hormones, such as estradiol (E2), regulates β-glucan-induced trained immunity responses in females, as shown in a septic mouse model via the non-canonical NF-κB pathway inhibiting the nuclear translocation of RelB [25]. These findings serve as a good basis to perceive the common understanding of males being more vulnerable to septic reactions than females and further offer a possible explanation about the sexual dimorphism existing in a variety of immune processes, such as individual responses to vaccines and pathogens, a process linked particularly to epigenetic changes occurring in the immune system [25,230,231]. On the contrary, an in vitro study by D’Avila et al., demonstrated that BCG training with increased production of pro-inflammatory cytokines can be suppressed by sex hormones [231,232]. However, the role of sex hormones altering the inflammatory state of innate immune cells remains controversial and needs further investigation.

Dietary mode serves as another important element that can govern the induction of trained immunity in innate immune cells [233]. Caloric restriction is known to improve health outcomes and longevity, hindering systemic inflammation driven by excessive energy intake and adiposity [233,234,235,236]. To date, reports have shown that the Western diet or the ketogenic diet, exclusively enriched in saturated fatty acids (SFAs), confers heightened responsiveness in myeloid cells upon a secondary challenge by endotoxins [237,238]. A NLRP3-dependent long-term functional reprogramming of precursor cells was reported as the main mechanistic event promoting trained immunity by dietary constituents [237]. Similarly, dietary components, such as bovine milk and milk-derived immunoglobulin G (IgG), drive the training effects on human monocytes in vitro upon a secondary insult by TLR ligands [239]. Altogether, these studies highlight the plasticity of innate immune cells to be reprogrammed by dietary supplementation types, resulting in the effective clearance of an infection or further potentially contributing to the severity of inflammatory diseases.

## 3. Endotoxin Tolerance

In contrast to the above-mentioned Schwartzman reaction, Beeson observed a progressive diminution of immune responses, Immune tolerance, in rabbits treated continuously by bacterial filtrates [53,240]. The phenomenon of ET was described as a key mechanistic development in myeloid cells regulating uncontrolled extensive inflammatory events, preventing harmful pathologies (i.e., sepsis) [30]. During ET, the cells enter into a temporary unresponsive state characterized by a decreased release of pro-inflammatory elements, such as TNF-α, IL-1 family of pro-inflammatory cytokines and IL-6 due to an impaired NF-κB translocation, as well as increased levels of the anti-inflammatory mediators, such as IL-10, TGF-β and arginase-1 (Arg-1) [29,30]. Inducible nitric oxide synthase (iNOS), as one of the main bacterial killing pathways, is also attenuated during ET [241]. A close competitive interplay of Arg-1 and iNOS regulating ET was reported since both enzymes share the common substrate L-arginine [242,243,244]. To date, numerous studies proclaim that, in addition to myeloid cells (i.e., monocytes, macrophages, dendritic cells, microglia and neutrophils), the development of ET also occurs in lymphoid cells, such as T cells, and non-immune cells (i.e., endothelial cells) [245,246,247,248,249,250,251,252].

The phenomenon of ET is a non-specific event, which is not limited to a specific receptor or element as the main reason for the cell anergy, since many cytokines, pathogens, fever or endogenous glucocorticoids can mimic the effects of endotoxin and LPS [253].

The activation of TLR4 by LPS recruits two distinct signaling routes, the MyD88-dependent and -independent (TRIF) pathways [254]. Both signaling events trigger the activation of NF-κB in a TRAF6 or TBK1-dependent manner, respectively [254]. However, during ET, most LPS-induced signaling events are impaired, but not entirely shut down since several anti-inflammatory mediators (i.e., IL-10) are upregulated—a process driven particularly by the delayed signaling of the MyD88-independent pathway [254,255]. TREM-1, as an orphan immunoreceptor, drives the inhibition of MMPs, also resulting in a diminished cytokine production thus contributing to the control of ET [47,256,257]. To date, researchers identified many negative regulators (SHIP, IRAK-M, SOCS1 and A20) that are implicated in the development of ET and innate immune homeostasis [254,258,259,260]. Furthermore, ongoing studies have demonstrated that the elevated release of IL-10 counteracts pro-inflammatory responses by sustaining the phosphorylation of Stat3 proteins [261]. AMPK serves as another well-known sensor and mediator of energy expenditure exerting the regulatory features of ET by suppressing LPS-induced TNF-α release and supporting anti-inflammatory reactions [196,197,262].

The transcriptomic and epigenetic analysis of myeloid cells undergoing ET have shown that this mechanistic process is accompanied by epigenetic modifications occurring on histones and regulated mainly by miRNAs. Histone modifications, especially deacetylation or methylation, cause gene silencing and are associated directly with the development of ET [30,263]. Tolerant human leukocytes display defective p65 activation on the promoter of IL-1β driven by the repressor transcription factor RelB via histone H3K9 dimethylation (H3K9me2) [35]. Comparable findings have been reported for human leukocytes, where the transcription silencing of acute pro-inflammatory genes (repression phase) during the severe systemic inflammation (SSI) is linked to chromatin remodeling, specifically to H3K9 histone methylation and RelB binding [264]. Likewise, both types of positive histone modifications, such as H3K4me3 and H4 acetylation, are selectively lost on the promoters of some pro-inflammatory genes, such as IL-6, during ET [34]. Further investigations have shown that gene-specific silencing by RelB is linked to the recruitment of histone H3 lysine 9 methyltransferase G9a and HP1 [35,36].

MiRNAs, as small non-coding RNAs (~22 nucleotides), were also described to be involved in the post-transcriptional regulation of inflammatory reactions during ET. Numerous studies described several miRNAs as being involved in the development of ET, such as miR-98, miR-125b, miR-132, miR-146a, miR-155, miR-221, miR-222, miR-579 and let-7 family [47,265,266]. For instance, miR-146, miR-155 and let-7 family target TLR4, attenuating signaling through the regulation of TRAF6 or IKKε, triggering downregulation inflammatory responses [267,268,269,270,271,272]. Diminished levels of TNF-α, as a result of subsequent challenge by LPS, were reported by the activation of miR-125b, targeting the 3′-untranslated regions of TNF-α transcripts [269]. MiR-132 has been shown to weaken inflammation in the brain, whereas miR-212 serves as a well-known tumor suppressor [273,274]. Both are known to be induced by TLR2 activation and being involved in cross tolerance responses through IRAK-4 modulation [275]. Lately, Seeley and colleagues identified miR-221 and miR-222 as critical regulators of macrophage reprogramming during LPS-tolerization, particularly associated with immunoparalysis and organ damage [276]. Moreover, miR-579 is also reported to reduce inflammatory reactions by inhibiting the translation of TNF-α in macrophages [249,277]. On the contrary, miR-98 targets the 3′-untranslated regions of IL-10 transcripts as a key cytokine for the development of ET; thus, the decreased activation of miR-98 in macrophages is accompanied with reduced levels of IL-10 [249,278]. However, the role of other miRNAs in the induction of endotoxin tolerance remains elusive and needs to be clarified. Furthermore, it would be of profound interest to determine whether the dysregulation of miRNAs may significantly affect the development and progression of inflammatory diseases [265].

Earlier studies disclosed weakened metabolic changes, especially affecting glucose metabolism during ET [279]. Changes in bioenergetics occurring during ET have been reported to be linked to several intermediate signaling proteins, such as Akt-1 or sirtuin-1 (SIRT1) [280,281]. A proper induction of ET by LPS requires the involvement of diverse intermediate proteins, such as Akt1 or SIRT1, anticipating the diminished levels of pro-inflammatory cytokines (i.e., TNF-α, IL-6 and IL-12p40) in mammals [282,283]. An increasing number of studies have suggested a close interplay between metabolism and inflammatory responses [284,285,286]. Impaired glycolysis is likely to be associated with declined inflammatory responses [287]. Supporting evidence has shown that decreased levels of lactate as an end product of glycolysis triggers reduced pro-inflammatory features and supports an anti-inflammatory phenotype in immune cells, especially on myeloid cells [288,289,290,291]. To date, reports postulated that LPS-tolerant innate immune cells are associated with reduced glycolysis as well as dysregulated cellular respiration, a process that alters the production of adenosine triphosphate (ATP) [166,211,292,293]. Recent work from Chen et al. demonstrated that the prominent gene immune-responsive gene 1 (IRG1) of activated myeloid lineage cells is responsible for encoding mitochondrial enzymes that drive the synthesis of itaconate, supporting anti-inflammatory responses mainly by modifying crucial glycolytic enzymes (impaired glycolysis) in LPS-stimulated cells [294,295]. At a mechanistic level, the attenuated inflammation by endogenous itaconate is attributed to the inhibitory effects on TET-family DNA dioxygenases, especially TET2 as a major target, thus suppressing the activation of NF-κB and STAT signaling routes [294,296]. Comparable findings for itaconate expressing anti-inflammatory properties were also highlighted in an earlier investigation by Lampropoulou et al., where they demonstrated that itaconate modulates the cell metabolism by suppressing the oxidation of succinate as a key regulator of the IL-1β–HIF-1α axis by succinate dehydrogenase [297].

Another metabolite that facilitates ET serves α-ketoglutarate (αKG). αKG is an end product of glutamine metabolism (glutaminolysis) that exerts anti-inflammatory properties of myeloid cells by modulating the NF-κB and Jmjd3 signaling pathways [298,299]. In addition to αKG, the engagement of fatty acid oxidation (FAO) has been shown as a crucial metabolic event contributing to the development of ET by AMPK-dependent activation [166,298,300,301,302,303]. Altogether, based on these characteristic features, we may conclude that ET, specifically cross-tolerance as a non-specific feature mainly attributed to innate immune cells, can serve as an antagonistic reaction to trained immunity.

## 4. Role of Trained Immunity and Endotoxin Tolerance in Inflammatory Diseases

Training as well as tolerance represent memory-like (adaptive) manners of the innate immune cells. Both opposing adaptive reactions of the innate immune system are of fundamental interest since they may represent a double-edged sword counteracting each other driving, this protecting or supporting the pathogenesis of a variety of inflammatory diseases. To date, the number of studies suggesting the involvement of training or tolerance in inflammatory diseases is continuously rising (Figure 4). In this review, we briefly discuss the role of these adaptive cues in several inflammation-related disorders.

To date, trained immunity has been described as a non-specific defense mechanism where innate immune cells primed by different microbial or endogenous molecular ligands confer protection against a broad spectrum of lethal pathologic infections [21,24,72]. On the contrary, the maladaptive activation of trained immunity triggers deleterious effects due to uncontrolled hyperactivation contributing to the severity of inflammatory pathologies [28,304]. Atherosclerosis, as an inflammatory disease, represents one of the main causes of coronary heart diseases, which is mainly driven by innate immune cells, especially monocytes and macrophages [305,306,307]. Recently, it was shown that trained monocytes and macrophages by oxLDL, aldosterone or hyperglycemia can contribute to the pathophysiology of atherosclerosis [26,163,308,309,310]. A similar correlation between BCG vaccination and increased aortic atherosclerosis in rabbits has been reported in the late 20th century [311].

Periodontitis, as a local inflammatory disease of oral mucosa, is associated with increased risk of certain systemic inflammatory diseases [312,313]. A recent work by Li et al., demonstrated that periodontitis-induced trained immunity in bone marrow exhibited increased severity when subjected to inflammatory comorbidities, such as rheumatic arthritis [304]. Furthermore, it is well-known that the pathogenesis of rheumatic arthritis as a chronic inflammatory disease affecting predominantly the joints is driven by several innate immune cells (i.e., monocytes, macrophages and dendritic cells) [314,315,316,317]. A close interplay between trained innate immune cells and the progression of rheumatic arthritis has been postulated, where the augmented inflammatory response by myeloid cells is controlled by epigenetic and metabolic changes mediated by the mTOR/STAT3 pathway [318,319,320,321]. Likewise, BCG-trained murine macrophages, characterized by an elevated cytokine production, can modulate fibroblast transformation involving T/B cells affecting the severity of systemic sclerosis (SSc) and inflammatory fibrotic disorders [322].

Systemic lupus erythematosus (SLE) is a chronic autoimmune disease, where monocytes, macrophages and neutrophils play a prominent role in the disease progression [323,324,325,326]. An increasing number of studies has shown that, in SLE patients, myeloid cells manifest a dominant pro-inflammatory phenotype [327,328,329,330]. Neutrophil extracellular trap (NET) formation has been shown to participate in the pathogenesis of SLE [331]. Interestingly, *C. albicans*-injected mice demonstrated an increased production of NETs linked to exacerbated disease progression, a memory response attributed presumably to trained neutrophils [332,333].

A growing body of evidence has shown that β-glucan-trained myeloid cells, especially monocytes, macrophages and neutrophils, promote anti-tumoral activities controlling tumor progression, a process associated with transcriptomic, epigenetic and metabolic reprograming [61]. The appropriate rewiring of myelopoiesis by β-glucan or nanobiologic compounds, such as MTP10-HDL, results in the increased ability of innate immune cells to penetrate the tumoral environment, exhibiting suppressive effects on tumor growth [63,66]. The combination of trained immunity activities with the current immune therapy by checkpoint inhibitors represents a more efficient way of cancer immune therapy. However, it is well-known that prolonged inflammation fuels the progression of neoplastic transformation, supporting tumor growth into highly aggressive entities [334,335,336].

Nonetheless, a close association of training effects by innate immune cells supporting the pathogenesis of other inflammatory/metabolic diseases, such as diabetes mellitus type 2, obesity and chronic obstructive pulmonary diseases (COPD), has been suggested [308,337,338,339,340,341,342,343].

The process of neuroinflammation is linked to the development of neurodegenerative diseases, such as Alzheimer’s disease, Parkinson’s disease and amyotrophic lateral sclerosis [344,345,346]. An exaggerated immune response, training, by microglia contributes to the pathological hallmarks of neurodegenerative disease in mouse models [46,208,347,348]. On the contrary, the development of immune tolerance in microglial cell by continuous challenge by LPS could be a mitigating factor for neuroprotective effects in the brain [46,208,349].

The tolerant phenotype of innate immune cells, especially monocytes and macrophages, is primarily linked to the progression of cancer and sepsis, and has been observed in other diseases, such as hepatic ischemia, cystic fibrosis, pancreatitis and acute coronary syndrome [30,47]. Cytokines and chemokines released by tumor-associated macrophages (TAMs) and tumor-associated neutrophils (TANs) express pro-tumoral properties by promoting the angiogenesis, invasion and metastasis, while suppressing the immune system [350,351,352]. In vitro, tumor cells trigger the deactivation of monocytes by IRAK-M-dependent upregulation marked by declined levels of pro-inflammatory cytokines, a characteristic of tolerance development against neoplasms that can justify the increased risk of infections in leukemia patients [353,354,355,356,357]. Moreover, considering it from the phenotypic aspect, it is important to mention that TAMs exhibit comparable features to tolerant macrophages in vivo [358,359,360].

Sepsis is a life-threating organ dysfunction caused by a dysregulated host response to infections [361]. It represents a highly inflammatory disorder mediated by the uncontrolled activation of the immune system [362,363,364]. The early phase of sepsis is characterized by a hyper-inflammatory state where the induction of ET is likely to contribute to protective effects supporting decreased inflammation and repairing mechanisms [365,366]. The development of ET during the late phases of sepsis is associated with increased mortality due to suppressed immune functions and organ failure [47,366,367,368]. These findings indicate a far more complex role of ET during sepsis.

Hepatic ischemic-reperfusion (I/R) injury is a pathophysiological process of liver surgery accompanied by oxidative stress triggering cell damage and inflammation [369,370,371]. The local inflammatory reaction is driven by the innate immune cells [372]. In a rat model, it was shown that endotoxin preconditioning results in diminished hepatic I/R injury probably via the IRAK-4-dependent downregulation of intra-organ TNF-α expression [373,374]. Similar findings were reported earlier by other groups for I/R injury of the heart and hemorrhagic shock [375,376].

Other disease development which is linked to inflammatory responses of myeloid cells includes the cardiovascular disorder of acute coronary syndrome (ACS) [47,377]. This represents a good example of the hetero-tolerance where several internal factors, DAMPs (i.e., HMGB1, hyaluronic acid or heat-shock proteins), during ACS prime circulating monocytes toward a tolerant phenotype, failing to respond to subsequent challenges by external pathogenic agonists, such as LPS ex vivo [47,377]. As a result, ACS patients can be more vulnerable to secondary infections due to this phenomenon.

Finally, one of the most critical avenues is the investigation of the impact of trained immunity and endotoxin tolerance on different diseases, particularly how these adaptive features contribute to the pathogenesis of various diseases and how these reactions can be approached as preventive or therapeutic targets. As mentioned above, great emphasis has been placed on the development of trained immunity-based vaccines to increase the immunogenicity of current anti-infectious vaccines to stimulate broader responses [104]. Other relevant areas where the modulation of trained immunity may be of crucial relevance include: sepsis, cancer, allergy, neurodegenerative diseases and other immune-mediated pathologies [21,378]. Similarly, endotoxin tolerance was initially considered as a relevant mechanism to counteract hyper-inflammation-mediated diseases. As discussed previously, the immuno-suppressive state of endotoxin tolerance is involved in numerous pathologies, such as sepsis, hepatic and renal ischemia, cystic fibrosis, acute pulmonary syndrome and cancer [30]. Initial attempts of the modulation of these adaptive manners included designing proper small-molecule agonists/antagonist that may act in different PRRs; however, since some receptors (i.e., TLR4) may trigger the development of both opposing reactions, this attempt may not be of ideal use. Promising data are expected to appear in the coming years involving epigenomic and transcriptomic modulations as crucial regulators of trained immunity and tolerance.

Taken together, both opposing memory-like inflammatory reactions (training and tolerance) may play a crucial role in the progression or suppression of various inflammatory diseases. To date, the role of both adaptive cues of innate immune system in other diseases remains elusive and needs to be clarified.

## 5. Conclusions

The identification of these antagonistic inflammatory cues, trained immunity and endotoxin tolerance, has greatly challenged overall considerations accumulated through the years regarding the “primitivity” of the innate immune cells. While trained immunity aims at pathogen elimination, it may contribute also to the progression of inflammatory diseases. On the contrary, tolerance may promote repairing/protective mechanisms against some inflammatory pathologies, but on the other hand, its incidence correlates with an increased risk to secondary infections. Both opposing reactions of the innate immune system represent a highly complex, multi-level number of events notably regulated by diverse alterations occurring on signaling pathways, chromatin modulations as well as metabolic rewiring. However, we hope that future investigations in regard to the role of training and tolerance in various diseases will pave the way for the development of novel strategies for preventive and therapeutic purposes.

## Figures and Tables

**Figure 1 biomedicines-11-00766-f001:**
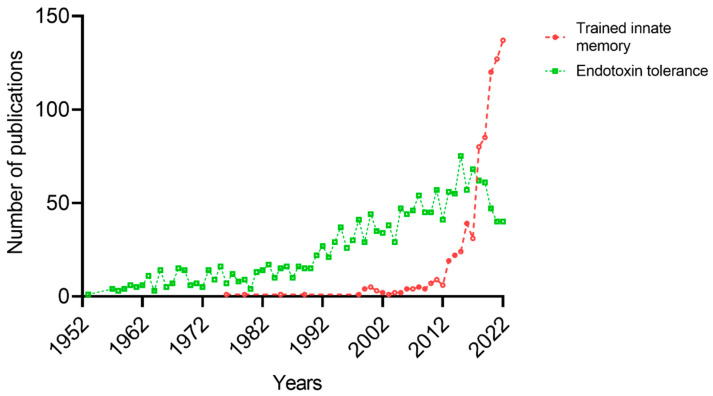
Number of publications related to trained innate memory and endotoxin tolerance as adaptive manners (PubMed, January 2023).

**Figure 2 biomedicines-11-00766-f002:**
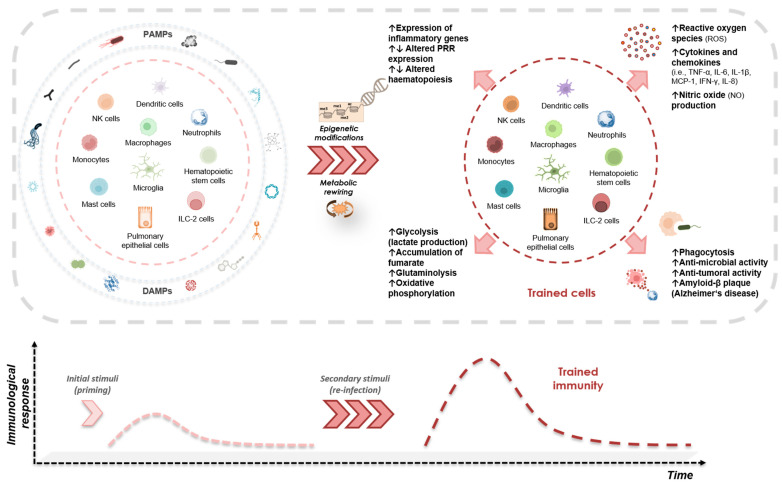
Stressor-dependent induction of trained immunity. Innate immune cells as well as non-immune cells primed by specific exogenous (i.e., PAMPs) and endogenous (i.e., DAMPs) signals evoke long-term non-specific functional modifications mediated by epigenetic and metabolic rewiring toward a secondary insult. Trained cells express a multi-level entity of features characterized by the increased production of inflammatory mediators and altered epigenetic and metabolic marks, and reflected functionally by changes occurring in several cellular activities.

**Figure 3 biomedicines-11-00766-f003:**
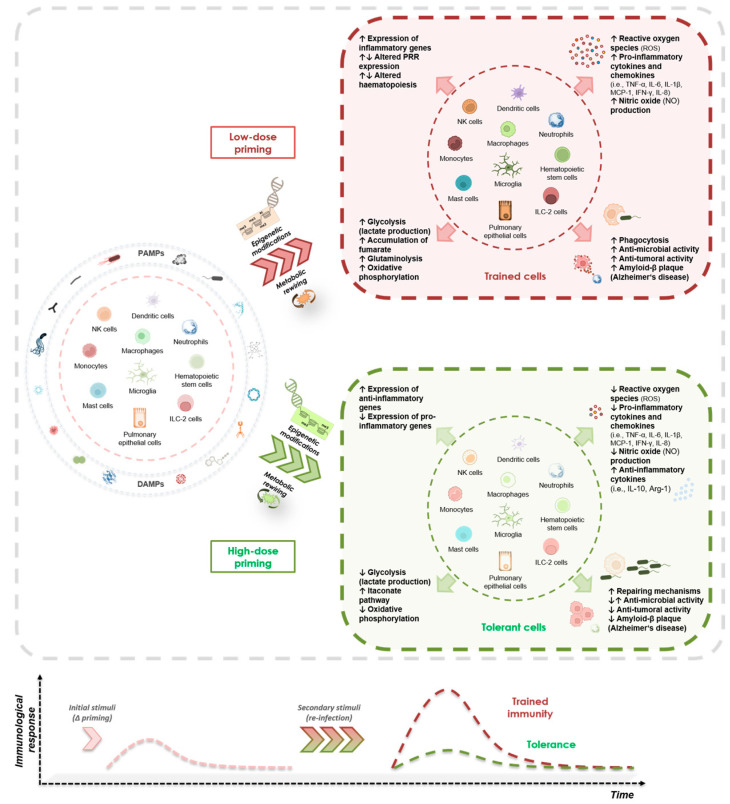
Dose-dependent induction of opposing adaptive inflammatory cues in innate immune system and non-immune cells. Innate immune cells primed by low/single dose PAMPs/DAMPs trigger a trained phenotype after subsequent encounter to a heterologous challenge, whereas high/repeated doses of PAMPs/DAMPs result in an immunotolerant phenotype. Both memory-like manners are mediated by epigenetic modifications with resulting effects on metabolism, altering various functional properties of the cells. This figure serves only for illustrative explanation of the hypothesis, since there are still missing data for such dose–response behaviors for certain types of cells.

**Figure 4 biomedicines-11-00766-f004:**
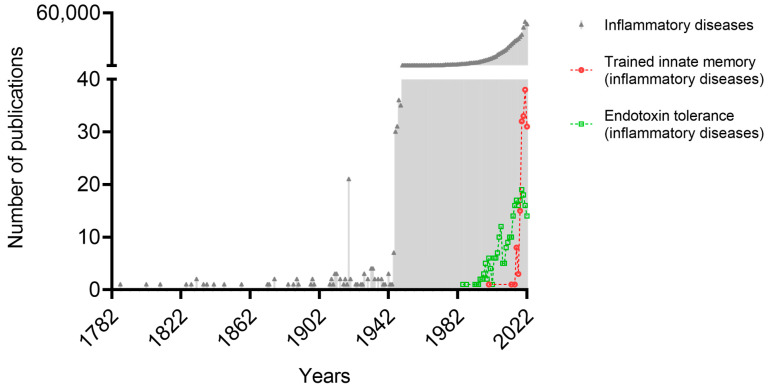
A timeline of the increasing number of publications related to the induction of trained immunity and endotoxin tolerance in various inflammatory disorders. Both adaptive inflammatory processes appear to exhibit a crucial role on the suppression or progression of several inflammatory diseases (PubMed, January 2023).

## Data Availability

The PubMed data presented in this paper are accessible online for any reader (Link: https://pubmed.ncbi.nlm.nih.gov/ (accessed on 10 January 2023)).

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
