# Peer review of "Training vs. Tolerance: The Yin/Yang of the Innate Immune System"

_biomedicines, 2023, doi:10.3390/biomedicines11030766_

Round 1
Reviewer 1 Report
The article written by Trim Lajqi and colleagues is interesting and quite comprehensive. The authors could add some figures on the mechanisms of action involved in immunity and tolerance, for example of TLRs. However, the authors should add more critical considerations, to bring new data and thoughts on such an interesting and current topic as this. The bibliographic data reported are quite recent. As a whole, the article should be revised by introducing new insights to deepen the state of the art. The authors could also analyze separately the roles of immune cells involved in the tolerance process, and contextualize them in a broader general framework, taking different physiological and pathological conditions as reference.
Author Response
The article written by Trim Lajqi and colleagues is interesting and quite comprehensive. The authors could add some figures on the mechanisms of action involved in immunity and tolerance, for example of TLRs. However, the authors should add more critical considerations, to bring new data and thoughts on such an interesting and current topic as this. The bibliographic data reported are quite recent. As a whole, the article should be revised by introducing new insights to deepen the state of the art. The authors could also analyze separately the roles of immune cells involved in the tolerance process, and contextualize them in a broader general framework, taking different physiological and pathological conditions as reference.
Response:
We thank the reviewer for the positive comments and suggestions. During the preparation of the manuscript we planned to include some relevant molecular mechanisms on both, trained immunity and endotoxin tolerance; however, we realized that this information has been elaborated properly by many studies and any attempt to illustrate the mechanistic interplay here would be redundant [1–6]. In our manuscript relevant molecular mechanisms implying epigenetic and metabolic changes have been discussed.
We further revised the manuscript by adding state of the art insights in regard to clinical applications of both adaptive features which may evolve in the near future (lines 742 - 759). Basic molecular mechanisms involved in the induction of both opposing reactions under physiological conditions are so far well described; however, the role of them in different pathologies remains elusive and requires extensive work to elucidate any possible preventive or therapeutic applications.
To our knowledge there are several reviews that discuss the role of specific immune cells involved in trained immunity or tolerance. Since the aim of this review was to give a broader aspect of both opposing adaptive features, molecular mechanisms with resulting epigenetic and metabolic changes and their role in different inflammatory pathologies, we consider that a cell-specific description will fall beyond our intension of a focused (compressive) review. Pathologic as well as physiological events include a far more complex interaction of various cells, and this may be impossible to be divided and discussed properly.
[1] Netea, M.G.; Domínguez-Andrés, J.; Barreiro, L.B.; Chavakis, T.; Divangahi, M.; Fuchs, E.; Joosten, L.A.B.; van der Meer, J.W.M.; Mhlanga, M.M.; Mulder, W.J.M.; Riksen, N.P.; Schlitzer, A.; Schultze, J.L.; Stabell Benn, C.; Sun, J.C.; Xavier, R.J.; Latz, E. Defining Trained Immunity and Its Role in Health and Disease. Nat. Rev. Immunol., 2020, 20, 375–388.
[2] Owen, A.M.; Fults, J.B.; Patil, N.K.; Hernandez, A.; Bohannon, J.K. TLR Agonists as Mediators of Trained Immunity: Mechanistic Insight and Immunotherapeutic Potential to Combat Infection. Front. Immunol., 2020, 11, 622614.
[3] van Leent, M.M.T.; Priem, B.; Schrijver, D.P.; de Dreu, A.; Hofstraat, S.R.J.; Zwolsman, R.; Beldman, T.J.; Netea, M.G.; Mulder, W.J.M. Regulating Trained Immunity with Nanomedicine. Nat. Rev. Mater., 2022, 7, 465–481.
[4] Biswas, S.K.; Lopez-Collazo, E. Endotoxin Tolerance: New Mechanisms, Molecules and Clinical Significance. Trends Immunol., 2009, 30, 475–487.
[5] López-Collazo, E.; del Fresno, C. Pathophysiology of Endotoxin Tolerance: Mechanisms and Clinical Consequences. Crit. Care, 2013, 17, 242.
[6] Fan, H.; Cook, J.A. Molecular Mechanisms of Endotoxin Tolerance. J. Endotoxin Res., 2004, 10, 71–84.
Reviewer 2 Report
The manuscript of Trim Lajqi et al. is dedicated to exciting topic – acquiring by innate immune cells features of the immune memory, which means an ability to respond more quickly and effectively to a repeated stimulus. Authors oppose this phenomenon of so called trained immunity to another type of altered response to secondary impact – tolerance. Authors summarize and discuss multiple evidences of the trained immunity found in various cell types and focus more precisely on the endotoxin tolerance phenomenon. Such consideration of the contrasting innate immune responses leading to long-term cell adaptation as well as considering stressor- or dose-dependent hypotheses emphasizes novelty and originality of the manuscript.
The manuscript is well structured and illustrated.
No big weaknesses were found in this work, and I would recommend this review for publication.
Minor concern:
Adaptive features of NK cells are described scanty. Cytokine-induced memory-like NK cells deserve a mention in the context of the manuscript, whereas CMV-associated adaptive NK cells are considered to be more close to T and B adaptive cells.
Author Response
The manuscript of Trim Lajqi et al. is dedicated to exciting topic – acquiring by innate immune cells features of the immune memory, which means an ability to respond more quickly and effectively to a repeated stimulus. Authors oppose this phenomenon of so called trained immunity to another type of altered response to secondary impact – tolerance. Authors summarize and discuss multiple evidences of the trained immunity found in various cell types and focus more precisely on the endotoxin tolerance phenomenon. Such consideration of the contrasting innate immune responses leading to long-term cell adaptation as well as considering stressor- or dose-dependent hypotheses emphasizes novelty and originality of the manuscript.
The manuscript is well structured and illustrated.
No big weaknesses were found in this work, and I would recommend this review for publication.
Minor concern:
Adaptive features of NK cells are described scanty. Cytokine-induced memory-like NK cells deserve a mention in the context of the manuscript, whereas CMV-associated adaptive NK cells are considered to be more close to T and B adaptive cells.
Response:
We thank the reviewer for the valuable evaluation and positive recommendation for publication. We also realized that adaptive features of memory NK cells have been little described by us in this manuscript and decided to further discuss this issue in detail (lines 210 - 218).
Reviewer 3 Report
This is an excellent review article about the memory function of the initiate immune systems. Th review is well written, structured with a few figures supporting the conclusion. Proofreading is necessary before publication. Some grammar or typos were noted, such as ...be taken under (into) consideration, contrary (on the contrary), and (A) growing body of evidence...
Author Response
This is an excellent review article about the memory function of the initiate immune systems. Th review is well written, structured with a few figures supporting the conclusion. Proofreading is necessary before publication. Some grammar or typos were noted, such as ...be taken under (into) consideration, contrary (on the contrary), and (A) growing body of evidence...
Response:
We thank the reviewer for the valuable evaluation and positive comments. Grammar or typos have been corrected properly.